# The Effect of Myofascial Stretching on Mechanical Nociception and Contributing Neural Mechanisms

Abigail W. Anderson, Arthur Soncini, Kaitlyn Lyons and William J. Hanney *

School of Kinesiology and Rehabilitation Sciences, University of Central Florida, Orlando, FL 32816, USA; abigail.wilson@ucf.edu (A.W.A.); ka027025@ucf.edu (K.L.)
* Correspondence: william.j.hanney@ucf.edu

**Abstract:** Myofascial stretching is often prescribed in the management of musculoskeletal pain. However, the neural mechanisms contributing to a decrease in pain are unknown. Stretching produces a sensation that may act as a conditioning stimulus in a conditioned pain modulation response. The purpose of this study was to compare immediate changes in pressure pain thresholds (PPTs) during a low-intensity stretch, moderate-intensity stretch, and cold water immersion task. A secondary purpose was to examine if personal pain sensitivity and psychological characteristics were associated with the responses to these interventions. Twenty-seven (27) healthy participants underwent a cross-over study design in which they completed a cold water immersion task, upper trapezius stretch to the onset of the stretch sensation, and a moderate-intensity stretch. A significant condition x time effect was observed (F (8,160) = 2.85, $p < 0.01$, partial eta$^2$ = 0.13), indicating reductions in pain sensitivity were significantly greater during a cold water immersion task compared to moderate-intensity stretching at minutes two and four. Widespread increases in heat pain threshold and lower pain-related anxiety were moderately correlated with the response to the cold water immersion task but not stretching. Moderate-intensity stretching may not elicit a conditioned pain modulation response possibly because the stretch was not intense enough to be perceived as painful.

**Keywords:** pain; muscle sensitivity; nociception; myofascial stretching





## 1. Introduction

Resistance and aerobic exercise may produce immediate reductions in the perception of pain, termed Exercise-Induced Hypoalgesia (EIH), through neural mechanisms. EIH effects are largest at the exercising muscle but are also observed over muscles that are not exercising, suggesting local and systemic effects of exercise [1,2]. Exercise may be perceived as painful, in particular during high-intensity training. Therefore, discomfort during exercise may engage natural pain inhibitory pathways activated by the application of a noxious stimulus, such as a conditioned pain modulation response [3]. Conditioned pain modulation is the "pain inhibits pain" phenomenon in which the application of two noxious stimuli may result in a natural inhibition of pain through a descending pain inhibitory pathway [4–8]. Conditioned pain modulation is a surrogate measure of the central nervous system's ability to activate endogenous pain inhibition and may be behaviorally assessed with psychophysical measures including the cold pressure test [9].

Conditioned pain modulation and EIH may share similar characteristics as both induce a secondary painful location (cold pressor test or exercise) that could inhibit pain at a primary site [10,11]. Therefore, it is reasonable that exercise may share similar descending inhibitory mechanisms as conditioned pain modulation [12–14]. Stretching is an additional rehabilitation intervention often prescribed in the management of painful conditions that may also produce hypoalgesia (stretch-induced hypoalgesia) [15] measured by an increase in the pressure pain threshold (PPT) at local and remote sites [16]. However, the reasons why stretching reduces pain remain unclear [17,18]. Stretching produces an uncomfortable

sensation that may be perceived as painful and, therefore, this intervention could potentially act as a conditioning stimulus in a conditioned pain modulation response.

Tolerance to the stretch sensation may be associated with an endogenous descending inhibitory response that improves muscle extensibility [19,20]. Greater tolerance to the stretch sensation is related to improvements in muscle extensibility rather than passive stiffness [21]. The relationship between tolerance to stretch sensation and improvements in range of motion is further demonstrated by studies manipulating sensory input prior to the intervention. In studies comparing the effects of self-massage with a therapy ball to TENS application during stretching of the ankle plantar flexor muscles, ankle range of motion increased the greatest after self-massage with a therapy ball plus stretching. Self-massage with therapy balls may be painful at times and reduce the H-reflex amplitude, allowing for a greater range of motion [22,23]. Furthermore, the stretch sensation is adaptable as regular stretching increases the participant's tolerance to the uncomfortable stretch sensation [24,25]. However, when stretching to the point of pain is applied over a period of four weeks, flexibility changes are due to a change in the perception of pain rather than properties of the muscle [26]. Collectively, this suggests that the stretching sensation is key to activating a mechanism that elicits improvements in range of motion. Given that stretching may be eliciting a conditioned pain modulation response, it becomes reasonable to directly compare the effects of a cold pressor test against stretching. If comparable changes in the PPT are observed between a cold pressor test and stretching, it is possible these interventions may share similar neurophysiological mechanisms of conditioned pain modulation.

A higher intensity stretch may be necessary to elicit this response as it may be uncomfortable. The effects of stretching on passive tissues, such as a tendons or ligaments, that oppose joint movement (passive torque measured with an isokinetic dynamometer) appear to depend on intensity with a higher intensity producing significant improvements [27]. Furthermore, stretching at a higher intensity may result in greater improvements in range of motion and muscle extensibility [28]. However, prior studies have not directly compared different intensities of stretching on pain outcomes. A higher intensity stretch may be perceived as painful and, therefore, elicit greater increases in the PPT. Identifying the appropriate intensity for reducing pain is important for prescribing the optimal dose and in the investigation of mechanisms potentially contributing to stretch-induced hypoalgesia. Conditioned pain modulation may be elicited with the application of a moderate-intensity noxious stimulus, and rehabilitation interventions (massage) may elicit comparable changes in pain sensitivity as a psychophysical paradigm (cold pressor test) for this mechanism [29]. Elucidating mechanisms contributing to stretching may improve accuracy in selecting interventions. Furthermore, individual characteristics of pain sensitivity and psychological factors influence the perception of pain and response to exercise. Although exercise can generate a higher pressure pain threshold, having a negative family environment and fear of pain lead to its decrease [30]. For pressure pain ratings, both fear of pain and mood disturbances were associated with generating a lower EIH [31].

Therefore, the first aim of this study was to compare immediate changes in the PPT between a cold water immersion task (conditioned pain modulation paradigm), a moderate-intensity stretch, and a low-intensity stretch. The second aim of the study was to examine baseline pain sensitivity and psychological factors associated with the response to each intervention for the total sample. This study was conducted in healthy participants.

## 2. Materials and Methods

### 2.1. Participants

Healthy participants between 18 and 60 years old were eligible for the study if they were not currently experiencing musculoskeletal pain. Participants were excluded for: having a history of a chronic pain condition, regularly taking prescription pain or anti-coagulant medication, contraindications to the application of ice, the presence of medical conditions known to affect sensation, or surgery/injury/fracture to the neck or arm within

the past six months. After consenting to participate, participants also underwent a blood pressure and Physical Activity Readiness Questionnaire Plus (PAR-Q+) screening to ensure their safety with participation in study procedures.

### 2.2. Experimental Procedures

Participants attended three sessions as part of a within-subject repeated measures study. As a brief overview, during the first session, participants completed pain-related psychological questionnaires, Quantitative Sensory Testing (QST), and a cold water immersion task. During the second and third sessions, participants completed a low- or moderate-intensity stretch of the upper trapezius in a counterbalanced order among participants with measurement of the pressure pain threshold (PPT). The University of Central Florida Institutional Review Board approved all study procedures (Study #5498) and the trial was prospectively registered on clinicaltrials.gov (NCT #05891353). All participants provided written informed consent to participate.

First, participants reported demographic factors (age, sex, race, ethnicity) on a standard intake form and completed pain-related psychological questionnaires as these factors can influence response to exercise [30,32]. Participants completed the Pain Catastrophizing Scale (PCS), Fear of Pain Questionnaire-9 (FPQ-9), Pain Anxiety Symptom Scale-20 (PASS-20), and the Brief Resilience Scale (BRS). The PCS measures pain catastrophizing with a higher score indicating a higher catastrophizing level [33]. The FPQ-9 measures fear of pain with higher scores indicating greater fear of pain [34,35]. The PASS-20 measures pain-related anxiety with higher scores indicating higher levels of anxiety. The BRS measures resilience with higher scores representing higher levels of resilience [36].

Next, participants underwent the following QST, as individual differences in pain sensitivity occur in healthy participants and may influence response to exercise-based interventions [37,38]. Participants completed the heat pain threshold (HPT), thermal temporal summation (TS), pressure pain threshold (PPT), and pressure pain tolerance (PPTol) in the same order each time. During the QST procedures, participants rated pain intensity with the 101-point Numeric Pain Rating Scale (NPRS), in which patients verbally rate their pain from 0 indicating "no pain" to 100 indicating "the most intense pain sensation imaginable".

HPT was delivered to the dominant forearm and upper trapezius as a measure of pain sensitivity local and distant to the site of the intervention with a $2 \times 1$ thermode attached to a TCS II from QST.Lab (Strasbourg, France). The thermode increased from 32 °C to a maximum of 50 °C at 1 °C per second. Participants pressed a response button when the sensation first changed from "comfortable warmth to slightly unpleasant pain". Two trials were conducted at each site with the average analyzed. Multiple trials were performed due to improved reliability. An Intraclass Correlation Coefficient (ICC), two-way random effects absolute agreement model indicated excellent reliability between trial one and two at the forearm (ICC = 0.91) and upper trapezius (ICC = 0.93).

Next, to assess temporal summation, the thermode was placed on the palmar surface of the hand and delivered a train of 10 heat pulses peaking at 49 °C with the desired inter-stimulus interval. Participants rated their "second pain" during each heat pulse with the NPRS, a C-fiber-mediated behavioral manifestation of the wind-up phenomenon. Temporal summation was calculated by subtracting the pain rating during the first pulse from the fifth pulse [39].

After completing thermal pain testing, mechanical stimuli were delivered with a computerized pressure algometer (AlgoMed, Ramat Yishai, Israel) with a 1 cm diameter rubber tip. The pressure algometer was applied to the dominant forearm and upper trapezius at a constant rate until the sensation changed from "comfortable pressure to slightly unpleasant pain" (PPT). Two trials of PPT were performed at each site with the average analyzed. Pressure pain tolerance (PPTol) was also examined. The algometer was applied to the dominant forearm and upper trapezius until the participants were "no longer able to tolerate the sensation". Two trials were performed with the average analyzed.

After completing all QST procedures, participants then completed the cold water immersion task consistent with a conditioned pain modulation paradigm [13]. The PPT was first assessed at the web space of the non-dominant foot for two trials. Participants then immersed their dominant hand into a bath of cold water set to 12° Celsius (ARCTIC Series Refrigerated Bath Circulator, Thermo Fisher Scientific, Waltham, MA, USA) [14]. Participants rated their pain intensity during the cold water immersion task every 15 s with the NPRS. Pain ratings were averaged for the analysis. After removing their hand, PPT was repeated on the non-dominant foot. Given that we were interested in comparing this condition to the effects of stretching, this sequence was repeated four times for a total cold water immersion time of four minutes and five PPT assessments. This protocol was based on the previously published protocol [29].

Participants then attended two additional sessions consisting of a low-intensity stretch or a moderate-intensity stretch in a counterbalanced order among participants. Given that the primary aim was to compare the effects of a cold water immersion task to stretching interventions, the intervention duration was based on the duration of the cold water task. Additionally, the outcome during the intervention (PPT applied to the web space of the foot) was based on the cold water immersion protocol. Our research team has successfully implemented this approach previously to compare the effects of different intensities of massage to a cold water immersion task [29]. Now, these methods were applied to stretching.

The PPT was applied to the web space of the non-dominant foot before and after each minute of the stretching intervention (5 total assessments), allowing for comparison of pain sensitivity changes between interventions. Two trials were performed during each testing time and the average analyzed. For the intervention, participants completed a stretch of the dominant upper trapezius for one minute, four times (4 total minutes), under the supervision of a licensed physical therapist. To stretch the upper trapezius, participants sat in a chair with their dominant hand holding the side of the chair. Participants were instructed to complete contralateral cervical lateral flexion followed by ipsilateral rotation. Overpressure with the non-dominant hand was provided by the participant as needed. Participants were instructed they should feel a gentle stretch along the upper trapezius and each participant provided verbal confirmation of this.

Participants were instructed to rate the intensity of the stretch from 0–100 with the following anchors: 0 = no stretch, 50 = moderate intensity, and 100 = maximal stretch intensity. Participants rated the stretch intensity every 15 s during the one-minute interval of the stretch and were instructed that this may be uncomfortable or painful. Stretch intensity was averaged across minute intervals. For the low-intensity stretch, participants were instructed to maintain a stretch sensation = 0/100 and instructed that this should not be uncomfortable or painful. For the moderate-intensity stretch, participants were encouraged to obtain a stretch sensation = 50/100 and instructed that this may be uncomfortable or painful at times. This protocol is consistent with our previously published studies in massage [29]. Participants were provided with verbal instructions throughout the stretch to maintain the desired stretch intensity.

### 2.3. Statistical Analysis

IBM SPSS Version 29 was used for all statistical analyses (IBM SPSS, Armonk, NY, USA). Descriptive statistics were obtained for the entire sample with data presented as mean ± standard deviation or percentage. As part of data preparation, the research team's a priori decision was to remove participants who exceeded the safety threshold (>1000 kPa) for pressure. The PPT applied to the foot during each intervention was also Z-transformed and the team's a priori decision was to exclude participants with Z-scores > 1.96.

The first aim of the study was to compare Immediate changes in the PPT between a cold water immersion task, a moderate-intensity stretch, and a low-intensity stretch. A mixed model ANOVA was examined for the within subjects factor of time (PPT applied to the foot between four, one-minute intervals of the intervention) x the between subjects

factor of condition (cold water immersion, moderate-intensity, low-intensity) effects. A simple effects decomposition with Bonferroni correction was performed.

The second aim of the study was to examine the correlation between baseline pain sensitivity and psychological factors with the response to each intervention. Changes in pain sensitivity were calculated for each intervention by subtracting the PPT applied to the foot before the intervention from the PPT applied to the foot after four minutes of the intervention. A negative value indicated hypoalgesia. Separate Pearson Correlation Analyses were performed for each intervention to determine the association between immediate changes in the intervention and psychological factors (FPQ-9, PASS, PCS, BRS) as well as pain sensitivity behaviorally assessed with QST during the first session (HPT, TS, PPT, PPTol). The magnitudes of associations were interpreted with the following thresholds for the correlation coefficient: 0.3 = weak, 0.5 = moderate, and 0.8 = strong [40].

## 3. Results

Thirty participants enrolled on the study; however, three individuals were removed from the data analysis. One participant did not complete the study, the second participant exceeded the safety thresholds (>1000 kPa), and the third participant demonstrated a Z-score > 1.96 (Z = 2.91). As a result, twenty-seven participants were analyzed. Results are presented as the mean $\pm$ standard deviation. Table 1 presents the demographic, pain sensitivity, and pain-related psychological characteristics for the total sample.

**Table 1.** Demographic, QST, and psychological characteristics for the total sample.

| Total Sample (n = 27) | Mean $\pm$ SD or % |
|---|:---:|
| Demographic characteristics | |
| Age (years) | 22.88 $\pm$ 4.76 |
| Sex (% female) | 70% |
| Race | |
| American Indian | 2.8% |
| Asian | 7.7% |
| Black | 7.7% |
| White | 80.8% |
| Ethnicity (% Hispanic) | 40.7% |
| QST characteristics | |
| HPT temperature forearm | 42.41 $\pm$ 2.64 |
| HPT temperature trapezius | 42.45 $\pm$ 2.93 |
| TS | 4.67 $\pm$ 11.39 |
| PPT forearm | 355.96 $\pm$ 233.50 |
| PPT trapezius | 346.40 $\pm$ 194.58 |
| PPTol forearm | 531.04 $\pm$ 213.68 |
| PPTol trapezius | 524.28 $\pm$ 202.66 |
| Psychological characteristics | |
| PASS | 26.25 $\pm$ 17.69 |
| BRS | 3.67 $\pm$ 0.56 |
| FPQ | 25.85 $\pm$ 4.97 |
| PCS | 13.81 $\pm$ 9.64 |

Note: HPT = heat pain threshold recorded in degrees Celsius, TS = temporal summation, PPT = pressure pain threshold recorded in kilopascals, PPTol = pressure pain tolerance recorded in kilopascals, PASS = Pain Anxiety Symptom Scale, BRS = Brief Resilience Scale, FPQ = Fear of Pain Questionnaire, PCS = Pain Catastrophizing Scale.

### 3.1. Changes in PPT between Interventions

Average pain intensity during the cold water immersion task = 39.05 $\pm$ 24.64. Participant-rated perceived stretch intensity = 27.87 $\pm$ 20.87 during the moderate-intensity stretching and 10.55 $\pm$ 13.39 during the low-intensity stretch. Mauchly's Test of Sphericity was not significant for condition x time effects ($p = 0.45$) and, therefore, sphericity assumptions were met. Mauchly's Test of Sphericity was significant for condition ($p = 0.02$) and, therefore, Greenhouse Geiser-corrected values are presented. Sphericity assumptions were met for the main effect of time ($p = 0.20$).

As demonstrated in Figure 1, a significant condition x time effect was observed (F(8160) = 2.85, $p < 0.01$, partial eta2 = 0.13). At baseline, low-intensity stretching displayed a significantly higher PPT over the foot compared to moderate-intensity stretching ($p = 0.02$, mean difference = 44.35 kPa, 95% CI: 6.79, 81.90). After two and four minutes of stretching, the cold water immersion task displayed a significantly higher PPT over the foot compared to moderate-intensity stretching (minute 2: $p = 0.01$, mean difference = 51.84 kPa, 95% CI: 9.13, 94.55; minute 4: $p = 0.03$, mean difference = 57.28 kPa, 95% CI: 5.58, 108.98).

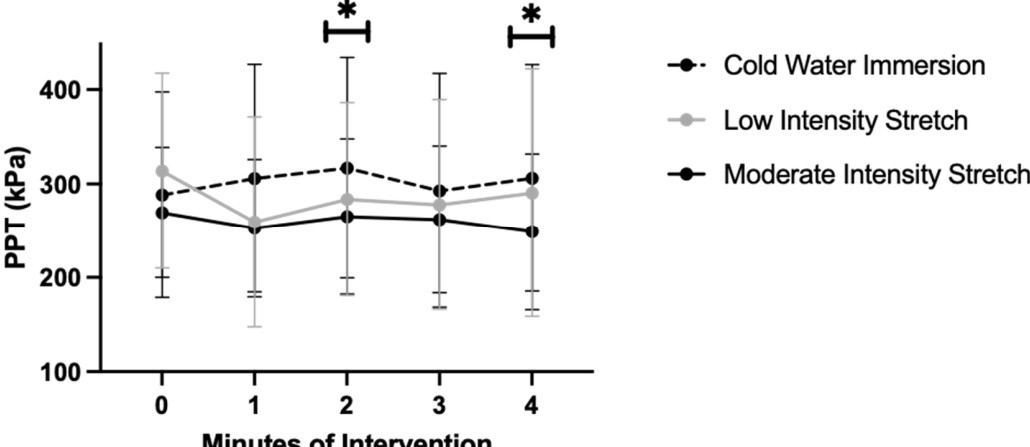

**Figure 1.** Pressure pain threshold by intervention. Note: * indicates statistical significance $p < 0.05$. Significantly higher PPT demonstrated during the cold water task at minutes two and four than the moderate-intensity stretch.

The main effect of condition (F (1.76,35.11) = 4.00, $p = 0.03$, partial eta2 = 0.17) was significant. The cold water immersion task trended toward displaying a significantly higher PPT compared to moderate-intensity stretching ($p = 0.05$, mean difference = 43.35 kPa, 95% CI: −0.69, 85.38). The main effect of time (F (4,80) = 1.73, $p = 0.15$, partial eta2 = 0.08) was not significant.

*3.2. Correlation among Baseline Factors and Response to Interventions*

Cold Water Immersion: As demonstrated in Table 2, a Pearson Correlation examined the data for associations between hypoalgesia observed during the cold water immersion task and baseline pain sensitivity factors (HPT, TS, PPT, PPTol). Inhibitory effects during a cold water immersion task displayed a significant moderate association with a higher HPT applied to the forearm (r = −0.49, $p = 0.01$) and trapezius (r = −0.46, $p = 0.02$). Thermal temporal summation displayed a small, non-significant association (r = 0.24, $p = 0.22$). The PPT and PPTol displayed no association with the response to this intervention (r range = −0.04−−0.17, $p > 0.05$).

For psychological factors, the PASS displayed a moderate, significant association with response to the cold water immersion task (r = 0.40, $p = 0.04$). This suggests that individuals with lower pain-related anxiety displayed greater hypoalgesia in response to a cold water immersion task. The FPQ-9, BRS, and PCS were not associated with the response to this intervention (r range = 0.04–0.06, $p > 0.05$).

Moderate-Intensity Stretch: Baseline pain sensitivity was not significantly associated with the response to a moderate-intensity stretch. Correlation coefficients demonstrated no association (r range = 0.06, −0.29, $p > 0.05$). Psychological factors displayed no association with the response to the intervention (r range = 0.03, 0.14, $p > 0.05$).

Low-Intensity Stretch: Baseline pain sensitivity was not significantly associated with the response to a low-intensity stretch. Correlation coefficients demonstrated no association (r range = 0.06, 0.31, $p > 0.05$). Psychological factors displayed no association with the response to the intervention (r range = 0.03, −0.17, $p > 0.05$).

**Table 2.** Association between response to intervention and baseline pain sensitivity, psychological factors.

| | Cold Water Immersion | | Moderate-Intensity Stretch | | Low-Intensity Stretch | |
|---|---|---|---|---|---|---|
| | **Pearson r** | ***p*-Value** | **Pearson r** | ***p*-Value** | **Pearson r** | ***p*-Value** |
| | | | QST | | | |
| HPT forearm | −0.49 | 0.01 * | −0.29 | 0.15 | 0.19 | 0.34 |
| HPT trapezius | −0.46 | 0.02 * | −0.15 | 0.47 | 0.19 | 0.33 |
| Temporal summation | 0.24 | 0.22 | 0.17 | 0.38 | −0.16 | 0.44 |
| PPT forearm | −0.21 | 0.29 | −0.29 | 0.15 | 0.38 | 0.05 |
| PPT trapezius | −0.11 | 0.59 | −0.01 | 0.96 | 0.11 | 0.58 |
| PPTol forearm | −0.04 | 0.85 | 0.21 | 0.33 | −0.10 | 0.66 |
| PPTol trapezius | −0.17 | 0.41 | 0.01 | 0.93 | −0.06 | 0.75 |
| | | | Psychological Factors | | | |
| PASS | 0.40 | 0.04 * | 0.01 | 0.95 | −0.22 | 0.29 |
| BRS | 0.06 | 0.77 | 0.02 | 0.91 | 0.01 | 0.95 |
| FPQ | 0.04 | 0.85 | −0.15 | 0.46 | 0.03 | 0.89 |
| PCS | 0.29 | 0.14 | 0.02 | 0.93 | −0.20 | 0.32 |

Note: HPT = heat pain threshold, TS = temporal summation, PPT = pressure pain threshold, PPTol = pressure pain tolerance, PASS = Pain Anxiety Symptom Scale, BRS = Brief Resilience Scale, FPQ = Fear of Pain Questionnaire, PCS = Pain Catastrophizing Scale. * indicates statistical significance.

## 4. Discussion

Endogenous modulation of pain may be behaviorally measured with a cold pressor task (cold water immersion of the hand). In this study, our research team applied cold water immersion similar to how a rehabilitation intervention may be prescribed (four, one-minute intervals) as a method to compare immediate changes in the PPT between stretching and endogenous pain modulation. Our results suggest the cold water immersion task after two and four minutes produces significantly greater endogenous modulation of pain compared to moderate-intensity stretching of the trapezius.

The primary purpose of this study was to compare immediate changes in the PPT during a moderate-intensity stretch, low-intensity stretch, or cold water immersion task. A significant lessening of pain sensitivity was observed at two and four minutes of a cold water immersion task compared to the stretching intervention. Given the significant difference between interventions, this suggests moderate-intensity stretching may not elicit a conditioned pain modulation response. It is possible the stretching sensation was not painful enough to elicit an endogenous pain inhibitory response, but future studies would need to examine this. The secondary purpose of this study was to examine the association between baseline psychological and pain sensitivity factors and the PPT response to each intervention. Furthermore, while baseline characteristics were not associated with the response to stretching, widespread increases in HPT and lower pain-related anxiety were moderately associated with the response to a cold water immersion task. Collectively, this suggests that the greatest hypoalgesia is observed during a cold water immersion task, likely due to a conditioned pain modulation response, compared to moderate-intensity stretching that did not elicit this endogenous pain modulation response.

The prior literature demonstrates that range of motion increases after introducing a painful stimuli, such as a cold pressor test or stretching to the point of pain [19,20]. In a cross-over study of nineteen healthy male participants, participants underwent pre/post assessment of the PPT (quadriceps, biceps, deltoid) in response to two repetitions of a thirty second static stretching intervention of the knee flexors or two minutes of cold water immersion (conditioned pain modulation) [19]. Stretching was completed until the sensation changed to pain (stretch tolerance). Participants also underwent an exercise and quiet rest condition. Significant condition x time interaction effects were not demonstrated for the PPT; however, interventions that elicited an endogenous pain modulation response (exercise, stretching, cold water immersion) by introducing a painful stimulus improved

tolerance to stretch [19]. In our study, we adopted a similar cross-over design but expanded upon these results by, now, examining if varying intensities of stretching impact immediate changes in the PPT. Additionally, participants were instructed to stretch to a moderate intensity that may be uncomfortable. Our results indicate that the inclusion of a painful stimulus (cold water) produces significantly greater increases in the PPT than moderate intensity. However, comparable changes in pain sensitivity were observed between stretches of low intensity and cold water immersion. Moving until the point at which a stretch sensation is first perceived elicited more favorable changes in pain sensitivity than moving until a moderate stretch sensation is perceived.

Additionally, we add to this body of literature by demonstrating that the intensity of stretch may impact immediate changes in pain sensitivity. Prior research examining intensity of stretching has primarily focused on immediate changes in range of motion [28]. High-intensity, long-duration stretching produces significant improvement in joint range of motion compared to a short-duration, low-intensity protocol [17]. Increasing the intensity of stretching may be beneficial for range of motion improvements; however, these beneficial effects may not be observed for pain outcomes. While the research team applied two intensities of stretching, there are additional considerations that may have impacted the lack of significant effects for stretching. First, it is possible the stretching protocol was not intense enough to elicit changes in the perception of pain. However, pain during stretching may present a barrier to participation in this intervention for patients with musculoskeletal pain and low-intensity stretching may be more acceptable to patients. Second, it is possible that local effects would have been observed, yet this was not examined. The PPT was assessed at a distant site to remain consistent with the conditioned pain modulation protocol. Stretching applied to the cervical region may have elicited changes in the PPT at the upper trapezius but this was not examined. Future trials would need to confirm this result in a larger clinical sample.

Pain is a complex phenomenon that is influenced by biopsychosocial factors. Systemic immunologic or central neurological adaptation alterations that occur in chronic pain may impact the perception of pain. While this trial was conducted in healthy participants, the results of this study may differ if the trial was repeated in individuals with persistent pain due maladaptive neuroplastic changes that may occur in this population. In a study comparing changes in conditioned pain modulation and temporal summation after submaximal aerobic exercise across three groups of individuals (rheumatoid arthritis, fibromyalgia, and healthy controls), different responses to exercise were observed by the group. Individuals with rheumatoid arthritis and healthy controls demonstrated improvements in temporal summation, while those with fibromyalgia did not, suggesting that alterations in central nervous system adaptations impact the EIH response [41]. Biopsychosocial factors including central neurological adaptation, impact pain perception, and response to exercise interventions.

In addition to examining immediate changes in the PPT during each intervention, a secondary aim was examining baseline pain sensitivity and psychological factors that may be associated with the response to each intervention. This is important to understanding the personal characteristics of patients that may impact who reports reductions in pain during stretching. Our results demonstrate baseline pain sensitivity and psychological factors were not associated with the response to stretching. This is consistent with a prior stretch-induced hypoalgesia study in which temporal summation on distal and local sites did not impact the response to stretching [33].

However, these factors were associated with the response to a painful cold water immersion task. Baseline pain sensitivity factors of HPT applied over the forearm and trapezius demonstrated a moderate, significant association with hypoalgesia during a cold water immersion task. The TS, PPT, and PPTol displayed no association with the response to this intervention (r range = 0.04, 0.14, $p > 0.05$). Pain sensitivity, including heat stimuli, predicts the conditioned pain modulation magnitude measured with a cold pressor test in healthy participants [42]. When a cold water task is repeated, we also demonstrate this

effect remains. Heat, but not pressure stimuli, may have been associated with this response as both activate cutaneous receptors rather than deep somatic structures.

The PASS displayed a moderate, significant association with the response to the cold water immersion task (r = 0.39, *p* = 0.04). This suggests that individuals with lower pain-related anxiety displayed greater hypoalgesia in response to a cold water immersion task. Individuals with a higher level of pain anxiety reported significantly greater pain intensity and unpleasantness, and lower pain tolerance during the cold pressor pain procedure [43]. This relationship may exist because participants with high anxiety report significantly more fear in response to the cold pressor test [44]. Given that the cold water immersion task is painful, it is possible this task may be impacted by pain-related psychological factors more than stretching. This is further supported by our results demonstrating psychological factors were not associated with the response to stretching. Collectively, personal characteristics may impact the response to a cold water immersion task but not stretching.

The limitations of this study include the sample size and short time frame of the study. We were able to analyze twenty-eight participants and were adequately powered to generate a significant interaction. However, future trials may aim to explore this in a larger sample of individuals with persistent pain. Furthermore, a longer duration trial may be more consistent with how this would be applied in clinical practice (over several sessions) and provide greater data on long-term effects. An additional limitation of the study is that it was conducted in healthy participants and, as a result, the effects of stretching on individuals experiencing pain is unknown. Because this study was conducted in healthy individuals, pain-related psychological factors may be reduced in this group. Therefore, they may not reach the threshold where psychological status would impact pain. This should be acknowledged as a limitation of the study and also that, while significant, the relationship between the PASS and cold water immersion may not be large enough to be clinically meaningful.

## 5. Conclusions

In conclusion, this study sought to examine the immediate changes in the pressure pain threshold (PPT) between a cold water immersion task, a moderate-intensity stretch, and a low-intensity stretch. Low-intensity stretching produced a significantly higher PPT after four minutes of stretching compared to moderate intensity. While pain sensitivity and psychological factors were not associated with this response, baseline HPT and PASS scores were correlated with the response to cold water immersion.

**Author Contributions:** Conceptualization, A.W.A., A.S., K.L. and W.J.H.; methodology, A.W.A. and A.S.; formal analysis A.W.A. and A.S.; data curation, A.W.A. and A.S.; writing—original draft preparation A.W.A., A.S., K.L. and W.J.H.; writing—review and editing, A.W.A., A.S., K.L. and W.J.H.; project administration, A.W.A. All authors have read and agreed to the published version of the manuscript.

**Funding:** This research received no external funding.

**Institutional Review Board Statement:** The study was conducted in accordance with the Declaration of Helsinki and approved by the Institutional Review Board of the University of Central Florida (protocol code 00005498).

**Informed Consent Statement:** Informed consent was obtained from all subjects involved in the study.

**Data Availability Statement:** The raw data supporting the conclusions of this article will be made available by the authors on request.

**Conflicts of Interest:** The authors declare no conflicts of interest.

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
