# Peer review of "The Effect of Myofascial Stretching on Mechanical Nociception and Contributing Neural Mechanisms"

_neurosci, doi:10.3390/neurosci5020011_

Round 1
Reviewer 1 Report
Comments and Suggestions for Authors
This is an interesting investigation looking at the effects of a cold stimulus or a cervical, stretching protocol on pain sensitivity/tolerance of the upper limb in healthy volunteers. The results indicate that a cold stimulus results in a greater hypo algesic effect compared to the stretching protocol. This topic is of interest to scientists and practitioners working with pain sciences/ patients, the manuscript is very well written and requires minimal grammatical editing.
there are a couple of issues that require adequate reply before
The introduction section, whilst well compiled, leaves a lot of assumptions of the reader in terms of the underlying scientific jargon and terminology. It is recommended that the authors where possible define key terms or provide one or two sentences on the background in certain areas for example, on line 29 ‘conditioned pain modulation’ is used, but the reader is not informed about the definition of this, nor the underlying scientific background
As highlighted by the authors pain is a complex phenomenon that has a immune basis as well as the perception of pain is influenced by psychological, social and cultural perspectives. So wholet the experimental trials were conducted in healthy volunteers. The nature of the pain stimulus is not necessarily of a clinical or pathological nature, whereby perhaps there is a systemic immunological response or central neurological adaptation that is impacting on that pain perception therefore, the study should be put in that context in terms of its applications, and the data that is presented. This should be better highlighted in the discussion section.
With regards to the stretching protocol. This is found to have of limited effect on the pain perception, and was concluded that the stretching protocol was not influential on a hypo algesia effect compared to the cold stimulus. However, was it that the stretching protocol was sufficient to induce any neurophysiological affect, especially when the stimulus was applied to the musculature itself.. At the same time stretching was appears to have been applied to the cerbicla region primarily region and not to the upper limb per se
Specific comments
Introduction
Line 27. What is the ‘distant sites’ that are referred to, then on line 29 what are the ‘descending inhibitory pathways’ please be more specific
Line 46. Please provide one line statement about the background to use of therapy balls and TENS as a pain modulating intervention
Line 59. What is meant by ‘joint passive torque’?
Methods llne, 124. What is the reliability given that two trials were conducted of the HPT?
Discussion
Please start the discussion section with a couple of sentences, summarising the important findings rather than a statistical statement
Lions, 310 to about 317 are good summarising statements, but also mentions the new additions to the literature. This should be your first statement of the discussion. Beyond that point listing the key results, then discussing with reference to previous research.
Online, 332 it is claims that psychological factors were not associated with the response to stretching. However, is it that the patients were healthy and there is a critical threshold where the psychological status of a patient will impact on pain, sensitivity, and these results. However, in this study that was not evident as all subjects were ‘healthy’ (Table 1 and description of subjects )
Even the results of PASS had a week correlation R=0.39 With the cold water immersion task which is not clinically relevant
The conclusion section should highlight some of these limitations, but also the important factor of an adaptive response to a cold stemless or the stretching protocol, which could present different results to a single session.
Author Response
Reviewer #1
Reviewer Comment 1: This is an interesting investigation looking at the effects of a cold stimulus or a cervical, stretching protocol on pain sensitivity/tolerance of the upper limb in healthy volunteers. The results indicate that a cold stimulus results in a greater hypoalgesic effect compared to the stretching protocol. This topic is of interest to scientists and practitioners working with pain sciences/ patients, the manuscript is very well written and requires minimal grammatical editing.
there are a couple of issues that require adequate reply before
Response to Comment #1: Thank you very much for your positive feedback! We hope this manuscript is useful to pain scientists and clinicians as this was our goal. We are happy to hear it was well written as well. We really appreciate your suggestions, in particular those related to the introduction and discussion section. We are extremely pleased with these improvements. Thank you again for taking the time to review our manuscript.
Reviewer Comment #2: The introduction section, whilst well compiled, leaves a lot of assumptions of the reader in terms of the underlying scientific jargon and terminology. It is recommended that the authors where possible define key terms or provide one or two sentences on the background in certain areas for example, on line 29 ‘conditioned pain modulation’ is used, but the reader is not informed about the definition of this, nor the underlying scientific background
Response to Reviewer Comment #2: This is an extremely good point and we are grateful to this reviewer for sharing their perspective on this. After reading the introduction again, we agree that there is a lot of scientific jargon and terminology. We have thoroughly reviewed the introduction for scientific terms and defined as appropriate. All changes can be found within the manuscript but we provided the specific edited section below for review:
“Resistance and aerobic exercise may produce immediate reductions in the perception of pain, termed Exercise Induced Hypoalgesia (EIH), through neural mechanisms. EIH effects are largest at the exercising muscle but are also observed over muscles that are not exercising , suggesting local and systemic effects of exercise. Exercise may be perceived as painful, in particular during high intensity training. Therefore, discomfort during exercise may engage natural pain inhibitory pathways activated by the application of a noxious stimulus, such as a conditioned pain modulation response. Conditioned pain modulation is the ‘pain inhibits pain’ phenomenon in which the application of two noxious stimuli may result in a natural inhibition of pain through a descending pain inhibitory pathway. Conditioned pain modulation is a surrogate measure of the central nervous system’s ability to activate endogenous pain inhibition and may be behaviorally assessed with psychophysical measures including the cold pressure test.”
Reviewer Comment #3: As highlighted by the authors pain is a complex phenomenon that has a immune basis as well as the perception of pain is influenced by psychological, social and cultural perspectives. So wholet the experimental trials were conducted in healthy volunteers. The nature of the pain stimulus is not necessarily of a clinical or pathological nature, whereby perhaps there is a systemic immunological response or central neurological adaptation that is impacting on that pain perception therefore, the study should be put in that context in terms of its applications, and the data that is presented. This should be better highlighted in the discussion section.
Response to Reviewer Comment #3: Thank you very much for this excellent suggestion. This is a good point to include that, while we tested in healthy participants, pain reflects alterations in varying biopsychosocial factors. To address this, we included a new paragraph in the discussion:
“Pain is a complex phenomenon that is influenced by biopsychosocial factors. Systemic immunologic or central neurological adaptation alterations that occur may impact the perception of pain. While this trial was conducted in healthy participants, results of this study may differ if the trial was repeated in individuals with persistent pain due maladaptive neuroplastic changes that may occur in this population. In a study comparing changes in conditioned pain modulation and temporal summation after submaximal aerobic exercise across three groups of individuals (rheumatoid arthritis, fibromyalgia, and healthy controls), different responses to exercise were observed by group. Individuals with rheumatoid arthritis and healthy controls demonstrated improvements in temporal summation while those with fibromyalgia did not, suggesting alterations in central nervous system adaptations impact EIH response. Biopsychosocial factors, including central neurological adaptation, impact pain perception and response to exercise interventions.”
Reviewer Comment #4: With regards to the stretching protocol. This is found to have of limited effect on the pain perception, and was concluded that the stretching protocol was not influential on a hypo algesia effect compared to the cold stimulus. However, was it that the stretching protocol was sufficient to induce any neurophysiological affect, especially when the stimulus was applied to the musculature itself.. At the same time stretching was appears to have been applied to the cerbicla region primarily region and not to the upper limb per se
Response to Reviewer Comment #4: We really appreciate this reviewer bringing this point to our attention and we agree that this should be better outlined within the discussion section. We have included the following:
“Additionally, we add to this body of literature by demonstrating the intensity of stretch may impact immediate changes in pain sensitivity. Prior research examining intensity of stretching has primarily focused on immediate changes in range of motion.21 High-intensity, long duration stretching produces significant improvement in joint range of motion compared to a short-duration, low intensity protocol.11 Increasing the intensity of stretching may be beneficial for range of motion improvements; however, these beneficial effects may not be observed for pain outcomes. While the research team applied two intensities of stretching, there are additional considerations that may have impacted the lack of significant effects for stretching. First, it is possible the stretching protocol was not intense enough to elicit changes in the perception of pain. However, pain during stretching may present a barrier to participation in this intervention for patients with musculoskeletal pain and low intensity stretching may be more acceptable by patients. Second, it is possible that local effects would have been observed yet this was not examined. PPT was assessed a distant site to remain consistent with the conditioned pain modulation protocol. Stretching applied to the cervical region may have elicited changes in PPT at the upper trapezius but this was not examined. Future trials would need to confirm this result in a larger clinical sample.”
Specific comments
Introduction
Reviewer Comment #5: Line 27. What is the ‘distant sites’ that are referred to, then on line 29 what are the ‘descending inhibitory pathways’ please be more specific
Response to Reviewer Comment #5: This is a good question and we appreciate the reviewer requesting to clarify this. Thank you very much for this suggestion. We have updated the sentence to the following:
“EIH effects are largest at the exercising muscle but are also observed over muscles that are not exercising, suggesting local and systemic effects of exercise. Exercise may be perceived as painful, in particular during high intensity training, that may activate shared inhibitory pathways, potentially including a Conditioned Pain Modulation response.”
Reviewer Comment #6: Line 46. Please provide one line statement about the background to use of therapy balls and TENS as a pain modulating intervention
Response to Reviewer Comment #6: We are happy to provide additional background on the use of therapy balls as a pain modulation intervention. We have updated this to the following:
“In studies comparing the effects of self-massage with a therapy ball to TENS application during stretching of the ankle plantar flexor muscles, ankle range of motion increased the greatest after self massage with a therapy ball plus stretching. Self-massage with therapy balls may be painful at times that reduce H-reflex amplitude allowing for greater range of motion.”
Reviewer Comment #7: Line 59. What is meant by ‘joint passive torque’?
Response to Reviewer Comment #7: We appreciate this reviewer suggesting that scientific jargon be minimized. This is a good suggestion to define this term. We have updated it to the following:
“Effects of stretching on passive tissues, such as a tendons or ligaments, that oppose joint movement (passive torque measured with an isokinetic dynamometer) appears to depend on intensity with higher intensity producing significant improvements.”
Reviewer Comment #8: Methods llne, 124. What is the reliability given that two trials were conducted of the HPT?
Response to Reviewer Comment #8: Thank you for this suggestion. We conducted a reliability analysis on the HPT trials and have included the following:
“Multiple trials were performed due to improved reliability. An Intraclass Correlation Coefficient (ICC), twoway random effects absolute agreement model indicated excellent reliability between trial one and two at the forearm (ICC=0.91) and upper trapezius (ICC=0.93).”
Discussion
Reviewer Comment #9: Please start the discussion section with a couple of sentences, summarising the important findings rather than a statistical statement
Response to Reviewer Comment #9: We greatly appreciate this reviewer’s suggestions on the discussion section as this has improve the readability. We have made several revisions to the discussion section based on this feedback. We have included the following summary statements at the start of the discussion section:
“The primary purpose of this study was to compare immediate changes in PPT during a moderate intensity stretch, low intensity stretch, or a cold water immersion task. A significant lessening of pain sensitivity was observed at two and four minutes of a cold water immersion task compared to the stretching intervention. Given the significant difference between interventions, this suggests moderate intensity stretching may not elicit a conditioned pain modulation response. It is possible the stretching sensation was not painful enough to elicit an endogenous pain inhibitory response, but future studies would need to examine this. The secondary purpose of this study was to examine the association between baseline psychological and pain sensitivity factors and PPT response to each intervention. Furthermore, while baseline characteristics were not associated with response to stretching, widespread increases in HPT and lower pain-related anxiety were moderately associated with response to cold water immersion task. Collectively, this suggests that the greatest hypoalgesia is observed during a cold-water immersion task, likely due to a conditioned pain modulation response, compared to moderate intensity stretching that did not elicit this endogenous pain modulation response.”
Reviewer Comment #10: Lions, 310 to about 317 are good summarising statements, but also mentions the new additions to the literature. This should be your first statement of the discussion. Beyond that point listing the key results, then discussing with reference to previous research.
Response to Reviewer Comment #10: Thank you for this suggestion. We moved this paragraph to the beginning. These changes can be found within the manuscript.
Reviewer Comment #11: Online, 332 it is claims that psychological factors were not associated with the response to stretching. However, is it that the patients were healthy and there is a critical threshold where the psychological status of a patient will impact on pain, sensitivity, and these results. However, in this study that was not evident as all subjects were ‘healthy’ (Table 1 and description of subjects )
Even the results of PASS had a week correlation R=0.39 With the cold water immersion task which is not clinically relevant
The conclusion section should highlight some of these limitations, but also the important factor of an adaptive response to a cold stemless or the stretching protocol, which could present different results to a single session.
Response to Reviewer Comment #11: Thank you for this suggestion. This is a good point that healthy individuals may inherently have less negative pain-related psychological factors that those with persistent pain. We agree these are important limitations to include within the manuscript. We have added this to the limitations of the paper:
“Because this study was conducted in healthy individuals, pain-related psychological factors may be reduced in this group. Therefore, they may not reach the threshold where psychological status would impact pain. This should be acknowledged as a limitation of the study and also that, while significant, the relationship between the PASS and cold-water immersion may not be large enough to be clinically meaningful.”
Reviewer 2 Report
Comments and Suggestions for Authors
The manuscript presented from Abigail W. Anderson et al., entitled "The effect of myofascial stretching on mechanical nociception" is interesting and original, however there are critical point to be adress:
- The authors reported only 24 reference (very low comparing with the wode number of reports)
- It is not clear the statistical analysis, the authors must improve the information concerning the analysis used, thy just write ANOVA, but it is not clear whish ANOVa. one-way? two-way? and why they choose it and not Nested ?
- The authors presented Figures without legends, it is really difficult to understand their results
- The authors should include raw data of all statistical analysis.
Author Response
Reviewer #2
Reviewer Comment #1: The manuscript presented from Abigail W. Anderson et al., entitled "The effect of myofascial stretching on mechanical nociception" is interesting and original, however there are critical point to be adress:
Response to Reviewer Comment #1: Thank you very much for your time reviewing this manuscript. We appreciate your feedback and have included responses within the manuscript and below.
Reviewer Comment #2: The authors reported only 24 reference (very low comparing with the wode number of reports)
Response to Reviewer Comment #2: Thank you for this feedback. In the original manuscript, we included 34 references. However, in these revisions, we have included additional references which has increased the total number to over forty references.
Reviewer Comment #3: It is not clear the statistical analysis, the authors must improve the information concerning the analysis used, thy just write ANOVA, but it is not clear whish ANOVa. one-way? two-way? and why they choose it and not Nested ?
Response to Reviewer Comment #3: Thank you very much for this suggestion and this has greatly enhanced our statistical analysis section. We selected this model because we had both between and within subject factors. Within subject factors were repeated measurements of PPT (defined as Time for the analysis) and the between subject factors were the condition (stretching and cold-water immersion tasks). A mixed model ANOVA is a combination of a between-unit ANOVA and a within-unit ANOVA. We did not select a nested model ANOVA because the groups were not subdivided into smaller groups. We have included the following:
“The first aim of the study was to compare immediate changes in PPT between a cold-water immersion task, a moderate intensity stretch, and a low intensity stretch. A mixed model ANOVA examined for within subjects factor of time (PPT applied to the foot between four, one-minute intervals of the intervention) x between subjects factor of condition (cold water immersion, moderate intensity, low intensity) effects. Simple effects decomposition with Bonferroni correction was performed.”
Reviewer Comment #4: The authors presented Figures without legends, it is really difficult to understand their results
Response to Reviewer Comment #4: Thank you for this suggestion. We have included the following description under the figure for added clarity:
“Note: * indicates statistical significance p<0.05. Significantly higher PPT demonstrated during the cold water task as minutes two and four than the moderate intensity stretch.”
Reviewer Comment #5: The authors should include raw data of all statistical analysis.
Response to Reviewer Comment #5: We appreciate this reviewer’s suggestion; however, submission of raw data is not required by the journal.
Thank you again for the opportunity to submit revisions of our manuscript.
Round 2
Reviewer 2 Report
Comments and Suggestions for Authors
The authors have satisfied all my concerns